# Comparing Synchronicity in Body Movement among Jazz Musicians with Their Emotions

**DOI:** 10.3390/s23156789

**Published:** 2023-07-29

**Authors:** Anushka Bhave, Josephine van Delden, Peter A. Gloor, Fritz K. Renold

**Affiliations:** 1MIT Center for Collective Intelligence, Cambridge, MA 02142, USA; bhaveanushka19@gmail.com (A.B.); jdelden@mit.edu (J.v.D.); 2Shanti Music Productions Renold & Co., 5012 Schönenwerd, Switzerland; fritzr@shanti-music.com

**Keywords:** facial emotion recognition, collective behaviour analysis, multi-person pose synchronization, convolutional neural networks, affective computing, pose estimation

## Abstract

This paper presents novel preliminary research that investigates the relationship between the flow of a group of jazz musicians, quantified through multi-person pose synchronization, and their collective emotions. We have developed a real-time software to calculate the physical synchronicity of team members by tracking the difference in arm, leg, and head movements using Lightweight OpenPose. We employ facial expression recognition to evaluate the musicians’ collective emotions. Through correlation and regression analysis, we establish that higher levels of synchronized body and head movements correspond to lower levels of disgust, anger, sadness, and higher levels of joy among the musicians. Furthermore, we utilize 1-D CNNs to predict the collective emotions of the musicians. The model leverages 17 body synchrony keypoint vectors as features, resulting in a training accuracy of 61.47% and a test accuracy of 66.17%.

## 1. Introduction

Emotion recognition is an important area of research to enable effective human–computer interaction [1]. Scientific research has led to applications of emotion recognition in tasks such as examining the mental health of patients [2], safe driving of vehicles [3], and ensuring social security in public places [4]. Collective emotions of a team refer to the shared emotional experiences and states that emerge within a group of individuals working together towards a common goal. These emotions are not just the sum of individual emotions but are experienced and felt collectively by the team as a whole. Collective behavior and group dynamics identify the synchronous convergence of an effective response across a group of individuals using data [5]. This multi-modal data may consist of facial configurations, textual sentiments [6], voice, granular data amassed from wearable devices [7], or even neurological data obtained from brain-computer interfaces [8]. Analyzing collective behavior aims to understand the emergent properties that arise from the interactions of individuals within a group [9]. These emergent properties may include collective intelligence, decision-making processes, flow, coordination, or conflict resolution [9]. The detection and analysis of emotions leveraging recent developments in artificial intelligence have seen progressive advancements using multi-modal datasets, machine learning, and state-of-the-art deep learning models. Understanding collective behavior and group dynamics is crucial for improving team performance [9]. By identifying factors that facilitate or hinder effective responses across the group, interventions can be developed to enhance collaboration, decision-making, and overall group performance [9].

Facial Emotion Recognition (FER) is a computer vision task aimed at identifying and categorizing emotional expressions depicted on a human face [10]. The goal is to automate the process of determining emotions in real-time, by analyzing the various features of a face such as eyebrows, eyes, and mouth, and mapping them to a set of basic emotions like anger, fear, surprise, disgust, sadness, and happiness [10]. Recently, researchers have turned to exploring emotions through body pose and posture, emotional body language, and motion [11]. The recent improvements in human pose estimation make pose-based recognition feasible and attractive [11]. Several recent studies propose further improvements in conducting body language prediction from RGB videos with poses calculated by OpenPose [12]. This study is also related to emotion recognition. Several previous studies propose to detect psychological stresses with multi-modal contents and recognize affects with body movements [13]. Unconsciously shared movement patterns can reveal interpersonal relationships: from the similarity of their poses, reciprocal attitudes of individuals can be deduced [13]. Estimation of body synchronization is relevant in a variety of fields like synchronized swimming [14], diving [15], and group dancing [16] which can profit from an analysis of motion and pose similarity. Organizational researchers focusing on leadership and team collaboration may be interested in studying human interactions through synchronization effects [17]. Psychological and sociological research is studying similar effects, too. For example, exploring the effect of body synchronization on social bonding and social interaction [18,19]. Interest in body synchronization stems from the objective to transfer inter-personal entanglement, a social network metric describing the relationship of individuals in their community, to human body movement. Bodily entanglement is defined as an overarching concept entailing the synchronization of bodies and their distance [20]. Entanglement as a social network metric has been proven to be an indicator of team performance, employee turnover, individual performance, and customer satisfaction [20]. The concept is based on earlier research that studied various forms of human synchronization, emotional body language, and activities that lead to a state of connection and flow between individuals [20].

Research on the estimation of body synchronization in a group of jazz musicians focuses on understanding how musicians coordinate their movements and actions during a performance. At the same time, we also look at how this is related to the overall flow, entanglement, and collective emotional behavior of the group. Glowinski, D. et al. [21] explore the automatic classification of emotional body movements in music performances using machine learning. Their study aims to develop computational models that can recognize and classify the emotions expressed through body movements. Participants performed musical tasks while their movements were analyzed and used to train machine learning algorithms. The results demonstrate the potential of automated systems to recognize affective body movements in music, with applications in affective computing and human–computer interaction. However, research on predicting the collective emotions of teams performing music using a quantified metric for body synchronization is lacking due to the limited availability of reliable tools for multi-person pose synchronization [22]. Existing tools are error-prone and tailored for specific purposes, hindering comprehensive studies [22]. Furthermore, there is a lack of integrated research, both technically and conceptually, examining the intricate bodily entanglement and flow of performing groups, such as jazz orchestras, to identify factors influencing group performance. Motivated by the aforementioned research problems, we inspect ways of examining the relationship between team entanglement and the collective emotions of a group of jazz musicians.

We study the data from a two-hour jazz rehearsal session performed by an orchestra of 19 musicians who were part of the Jazzaar festival (www.jazzaar.com). Figure 1 depicts musicians playing diverse instruments during the Jazzaar experiment. The chief contributions of the research presented in this paper are:We developed a high-performing system for real-time estimation of multi-person pose synchronization, detecting body synchronization across diverse visual inputs to calculate synchronization metrics. It leverages Lightweight OpenPose [23] for efficient pose estimation, achieving a performance of 5-6 frames per second on a regular CPU. By analyzing pre-recorded rehearsal videos of jazz musicians, we extract 17 body synchronization metrics, encompassing arm, leg, and head movements. These metrics serve as features for our deep learning model. The system incorporates a robust synchronization metric, enabling accurate detection across various pose orientations.To assess the relationship between facial emotions and team entanglement, we compute the Pearson correlation between facial emotions and various body synchrony scores. Additionally, we conduct a regression analysis over the time series data, using body synchrony scores as predictors and facial emotions as dependent variables. This approach allows us to estimate the impact of body synchrony on facial emotions, providing deeper insights into the connection between team dynamics and emotional expressions.We propose a machine learning pipeline to predict the collective emotions of jazz musicians using body synchrony scores to achieve accurate and interpretable results.

Our software differs from existing tools through its real-time capability, reliability in multi-person pose synchronization, and flexibility in accepting a wide range of inputs. We also leverage 1D CNNs which are particularly well-suited for processing sequential data, such as time series data [24], and can capture local patterns and dependencies within a sequence. The use of convolutional filters also extracts meaningful patterns, which reduces the need for manual feature engineering.

## 2. Related Work

### 2.1. Emotions

Psychologists have developed multiple frameworks to understand and classify human emotions. When it comes to distinguishing one emotion from another, researchers take two different perspectives [25]. The first perspective suggests that emotions are distinct and fundamentally different constructs. The second perspective argues that emotions can be characterized along a continuum or dimension. Paul Ekman, a renowned psychologist, is a key proponent of the former. He identifies six primary emotions: anger, disgust, sadness, happiness, fear, and surprise. According to this theory, other complex emotions can be derived from these fundamental emotions [26]. Another model known as Plutchik’s wheel of emotions presents eight core emotions: joy, trust, fear, surprise, sadness, disgust, anger, and anticipation [27].

The Circumplex model, following the continuum model, was developed by James Russell [28]. It presents a dimensional perspective on emotions. According to this model, emotions are organized in a circular space that encompasses two dimensions: arousal and valence. Arousal is represented along the vertical axis, whereas valence is depicted along the horizontal axis. The center of the circle represents a state of neutral valence and moderate arousal. Within this model, emotional states can be positioned at various levels of valence and arousal, or at a neutral level for one or both of these dimensions. The Circumplex model is commonly utilized to examine emotional responses to stimuli such as words that evoke emotions or facial expressions conveying emotions.

Emotions are commonly expressed through various modes, including language, voice or tone of speaking, physiology, and facial expressions [29]. Extensive research supports the notion that both speech data and facial expressions serve as strong indicators for accurately predicting emotions [30]. Additionally, physiological changes in the body can serve as important indicators of emotions. They emphasize the significance of physiological cues, such as heart rate, skin conductance, pupil waves, and hormonal responses, in assessing and understanding emotional experiences [31].

For instance, Li M. et al. [32] focus on using the pupil wave as a physiological signal for depression and anxiety assessment because pupil dilation is known to be associated with emotional arousal and cognitive processes. When an individual experiences different emotions, their pupils can dilate or constrict in response to changes in the autonomic nervous system and the release of neurotransmitters. Roessler J., et al. [33] propose a novel application of emotion recognition using physiological signals through the “Happimeter” smartwatch. By tracking changes in body signals, such as acceleration, heartbeat, and activity, the smartwatch predicts individual emotions with high accuracy. They conduct an experiment over three months in the innovation lab of a bank with 22 employees to measure individual happiness, activity, and stress. Their research showcases the potential of using physiological signals for real-time emotion recognition and its impact on promoting well-being and positive behavior. In a previous study, an electrical device worn by candidates on the chest measured temperature, location, sound, and body acceleration for sentiment analysis [34]. They experimented in multi-class emotion prediction using heart rate and virtual reality stimuli, to investigate if the heart rate signals could be utilized to classify four-class emotions. They used common classifiers like Support Vector Machine (SVM), K-Nearest Neighbours (KNN), and Random Forest (RF) to predict emotions. An emotion classifier was built using Decision Tree (J48) and IBK classifiers by collecting the data on blood volume pulse, galvanic skin response, and skin temperature attaining an accuracy of 97% [35].

By considering a combination of language, voice, physiology, and facial expressions, researchers can gain deeper insights into the complex and multi-faceted nature of human emotions.

### 2.2. Facial Emotion Recognition (FER)

Emotion recognition models have extensively utilized traditional machine learning algorithms. Researchers such as Mehta, D. et al. [36] employed Support Vector Machine (SVM), K-Nearest Neighbours (KNN), and Random Forest (RF) algorithms to achieve intensity estimation and emotion classification. Happy, S. et al. [37] implemented a facial emotion classification algorithm that combined a Haar classifier for face detection with Local Binary Patterns (LBP) histograms of various block sizes as feature vectors. These features were then classified using Principal Component Analysis (PCA) to identify six basic human expressions. Geometric feature-based facial expression recognition was explored by Ghimire, D. et al. [38] who identified 52 facial points as features. These features were fed into a multi-class AdaBoost and Support Vector Machine (SVM) system, achieving high recognition accuracy. Advancements in emotion classification research have seen the integration of deep learning techniques, including Convolutional Neural Networks (CNN) and Long Short Term Memory (LSTM), a type of Recurrent Neural Network (RNN). Jung, H. et al. [39] proposed a deep learning approach where one deep network extracted temporal appearance features from image sequences, whereas another deep network focused on temporal geometry features from facial landmark points. Jain, D. et al. [40] combined LSTM and CNN networks in a multi-angle optimal pattern-based deep learning method to label facial expressions. In a different study, Kahou, S. et al. [34] discovered the superiority of a hybrid CNN-RNN model over a standalone CNN for facial emotion recognition when leveraging multiple deep neural networks for different data modalities. Bhave, A. et al. [41] construct a model using XGBoost to predict the collective facial emotions of a group of jazz musicians using electrical signals generated by plants kept in the vicinity. Thus, the field of emotion classification has witnessed the incorporation of advanced deep learning techniques, demonstrating their efficacy in capturing nuanced emotional information from facial expressions. These approaches offer promising avenues for improving the accuracy and complexity of emotion recognition models.

## 3. Methodology

### 3.1. Extracting FER Time Series Data

In this experimental study, we have utilized the face emotion recognition (FER) algorithm developed by Page, P. et al. [42]. The real-time output of this algorithm is depicted in Figure 2, showcasing its effectiveness. To achieve facial emotion recognition, we have employed the faceapi.js JavaScript API (available at https://justadudewhohacks.github.io/face-api.js accessed on 15 February 2023). This API, built on top of the TensorFlowJS core API, enables face recognition directly within the web browser. The faceapi.js API consists of two separate neural networks, one for face detection and another for face expression and emotion recognition. During the experiment, the FER algorithm receives frozen frames per second from the video window, capturing the feeds from all participants’ cameras. For each detected face, the model calculates the average probabilities for the current emotion every second. This process involves assigning a probability between 0 and 1 to each recognized emotion, including neutral, happy, sad, angry, fearful, disgusted, and surprised. For instance, a label like “angry (0.8)” indicates that the particular face appears 80% angry and has a 20% likelihood of being annoyed or disgusted, as predicted by the model.

To determine the overall audience emotion score, we calculate the mean face emotion value from all the detected faces. This approach allows us to obtain a comprehensive understanding of the collective emotional state of the audience. By leveraging the FER algorithm and faceapi.js, we are able to analyze facial expressions in real-time, assigning probabilities to various emotions and deriving an aggregate measure of audience emotion. This detailed assessment facilitates a deeper exploration of the emotional dynamics within the audience and enhances our understanding of their emotional responses.

### 3.2. Team Entanglement

Entanglement is a metric that measures the synchronization of communication in work settings. It focuses on the similarity of communication patterns among team members, indicating the extent to which they communicate in a synchronized rhythm. The research on entanglement is motivated by the discovery that synchronized physiological signals between team members can enhance performance, as well as the concept of flow state, where intentions are synchronized and actions are harmonious [43,44]. In the organizational context, entanglement quantifies the flow state and synchronization in team communication, considering the internet-mediated interpersonal synchronization [45]. It uses the similarity of communication time series, such as emails, as a proxy for synchronization, while also considering the distance between individuals involved in the activity.

Gloor, P. et al. [20] suggest the study of entanglement via body measures. The two overarching components of bodily entanglement are synchronization and distance. Entanglement describes the synchronization of communication but also compares the distance of actors’ network positions to derive information on their interweaving. Synchronization itself is measured in terms of the difference between variables describing actor behavior. Distance defines the difference in the actors’ positions in the network. As synchronization is measured through the difference in a certain behavior of actors, a mapping to the human body is proposed by comparing the postures of individuals via a pose similarity metric. The distance measure, which compares the actors’ positions in the social network, is proposed to be transferred by studying the difference in their physical positions in the room. Defining bodily entanglement is a possible variation of entanglement; it can be composed of bodily synchronization and distance. Body entanglement is loosely defined as the set E of body part synchronization $S_{b}$ and body distance *d*.
(1)E={S,d}

Gloor, P. et al. [20] explore the effect of entanglement on team performance in the context of social networks. A hypothesis of the effect of bodily entanglement on team performance is derived. Figure 3 summarizes the research model and explains how this hypothesis is derived.

### 3.3. Real-Time Estimation of Multi-Person Pose Synchronization

We create a dedicated software measuring entanglement and body pose synchronization [22]. The software generates synchronization values for an individual body part. The software is a real-time multi-person pose synchronization estimation system, designed to be user-friendly, intuitive, and fast in execution. To address the limitations of previous synchronization measures and encompass a broader understanding of synchronization, four different types of synchronization metrics are made available in this software. These metrics aim to incorporate perpendicular body part differences into the synchronization score and provide the option to compare opposite-body sides, considering mirroring as a form of synchronization. The system’s implementation adheres to the designed framework and employs an analysis pipeline for each input frame. This pipeline includes pose estimation, derivation of pose synchronization, body distance, and body height for the current frame. At the end of the system run, tabular and visual output is generated and provided to the user.

During the implementation process, Lightweight OpenPose [23], an efficient pose estimation model, is used. The system achieves an average frame rate of 5.5 fps on a Mac M1 8-core CPU. It incorporates four synchronization metrics and offers two different output formats. The design of the system is visualized in Figure 4. The exemplary visual output generated by the software for input of recorded videos of jazz rehearsal is shown in Figure 5. We further describe the three major components implemented in the system.

#### 3.3.1. Pose Estimation

In this system, pose synchronization analysis relies on pose estimation, which refers to the process of determining the positions of human body joints using visual data, such as images or videos [46]. The system implements Lightweight OpenPose, which runs inference with 28 frames per second on a device with CPU access [23]. Lightweight OpenPose follows a bottom-up approach to pose estimation. The bottom-up approach involves initially detecting all the joints in an image, without prior knowledge of the number of individuals. Once all the keypoints are detected, they are then grouped based on the individuals present. In contrast, the top-down approach works in the opposite direction by first identifying the person instances in the input image. Subsequently, pose estimation is carried out for each individual instance [47]. Figure 6 [48] illustrates these differences.

The bottom-up approach proves to be more effective than the top-down approach when dealing with input images that have significant occlusions and complex poses. However, it tends to generate more false positives since it does not leverage body structure information. On the other hand, top-down approaches make better use of this information but struggle with complex poses and crowded scenes [49]. Images that pose challenges for top-down implementations, for instance, may exhibit characteristics such as pronounced torso rotation, cluttered backgrounds, or loose clothing that obscures body outlines [50]. A notable advantage of the bottom-up approach, which justifies its application in the developed system, is its resilience to the number of people present in the input image. Although top-down approaches require separate passes through the network for each person instance, resulting in increased execution time proportional to the number of individuals, bottom-up approaches perform a single pass on the input image. As mentioned earlier, the network outputs the overall image keypoints, and subsequent grouping of keypoints per person is conducted without the need for additional forward passes. Therefore, the processing time only marginally increases with higher person quantities [51].

Based on the estimated body joints, we refer to body parts as the vector between two adjacent keypoints, as listed in Table 1. The detected joints, or keypoints, are further illustrated in Figure 7.

#### 3.3.2. Synchronization Calculation

The system provides two types of mapping—linear and perpendicular. In the perpendicular approach, a 90° angle between body part vectors indicates a complete absence of synchronization, whereas angles approaching zero degrees indicate an improvement in synchronization. Angles nearing 180° also signify an enhancement in synchronization. In contrast, a linear synchronization variant interprets an angle of 180° as a complete lack of synchronization, with synchronization improving as the angle approaches 0°. The system encompasses four synchronization metrics as demonstrated in Figure 8 that are defined based on two key axes: the point of minimal synchronization: 90° angle or 180° angle and the body parts being compared: same-side or opposite-side as shown in Figure 9. The first axis focuses on how the angle between body part vectors is translated into a synchronization score, where the score can be zero at either a vector angle of 90° or 180°. The second axis distinguishes synchronization based on whether it considers same-side body parts or opposite-side body parts. This classification allows the system user to choose the most suitable metric for their specific use case, ensuring flexibility and adaptability.

#### 3.3.3. Distance

The pose distance in the system is determined by using the hip centers of the poses as reference points. The hip center would be the center between the two keypoints L-Hip and R-Hip, as shown in Figure 7. This approach yields a more stable time series compared to using reference points located on the limbs, which tend to exhibit higher degrees of movement. By calculating the Euclidean distance between the hip centers and multiplying it by the normalization factor ’H’, which is the sum of pose-independent body heights *’h’*, we obtain the distance between poses.

### 3.4. Data Extraction and Pre-Processing

The raw dataset of facial emotions comprises time series data for all seven emotions, namely *angry, sad, disgusted, neutral, happy, surprised,* and *fearful*. Each emotion is represented as a probability score ranging from 0 to 1, reflecting the likelihood of that emotion being expressed collectively at each second. For instance, a single data point in the dataset might indicate the following emotion probabilities: *angry* = 0.674142, *happy* = 0.152293, *sad* = 0.118742, *neutral* = 0.027505, *disgusted* = 0.007242, *surprised* = 0.019037, and *fearful* = 0.001040. In this example, *angry* has the highest probability, indicating that the dominant collective emotion for that specific moment would be labeled as *angry*. To obtain the Y column of our dataset, we further ascertain the dominant collective emotion for each data point and assign a label accordingly. For our data, we observe that the Y column consists of the labels *happy*, *sad*, and *angry* since these are the dominant collective emotions. To extract the group synchronization features, we employ the software discussed in Section 3.3, which provides us with a comprehensive set of 17 body synchrony metrics. Within this software, we are offered a selection among four distinct synchronization metrics. Among these options, we opt for the perpendicular variant, as it is purported to offer superior outcomes compared to the linear variant. Given the context of our analysis, wherein all the musicians are oriented toward the audience during their performance, it is worth noting that the notion of opposite-side synchronization would only hold significance if two individuals were engaged in a face-to-face interaction, such as a conversation. Thus, the same-side variant emerges as the most suitable for our purpose. The body synchrony scores include keypoint vectors represented as values per second, which constitute the ’X’ component of our dataset. After data cleaning and feature scaling, an imbalance is observed in the classes present in the Y column. The original data comprises 54% happy, 42% sad, and 4% angry samples.

We use min–max normalization to linearly rescale each feature in the dataset to map the original feature values into a specified bounded range of 0 to 1. This preserves the relative ordering of data points while aiding numerical stability and convergence in the machine-learning algorithm.

To address this bias, we utilize SMOTE (Synthetic Minority Oversampling Technique) [45] to generate synthetic samples, resulting in a balanced dataset with each label representing 33.3% of the samples.

SMOTE generates synthetic samples for the minority class by interpolating feature vectors between existing minority class examples and their k-nearest neighbors in the feature space, mathematically producing new data points that lie on the line segments connecting the original instances. We use cubic spline interpolation to approximate a smooth curve between data points by fitting cubic polynomials in each interval that ensures continuity of the first and second derivatives at the data points, providing a piece-wise continuous and differentiable curve that minimizes interpolation errors.

Additionally, we employ Stratified K-Fold Cross Validation (K = 3) to obtain train and test data splits, which aids in preventing overfitting, reducing bias resulting in the development of a generalized model.

## 4. Results

### 4.1. Correlation Analysis

We carry out a correlation analysis using the Pearson correlation coefficient comparing facial emotions and body synchrony scores.

Pearson correlation is a statistical measure that quantifies the linear relationship between two variables by computing their covariance divided by the product of their standard deviations. Section 5 also mentions significance values that indicate the probability of obtaining this correlation by chance alone, derived from a hypothesis test.

We observe an overall negative correlation between the facial emotions of disgust, surprise, anger, fear, and sadness and the body synchrony scores. The value of N for these correlations is 7948. In Section 5, we further discuss and interpret the correlations showcased in Figure 10.

### 4.2. Regression Analysis

We perform a regression analysis on the independent variables of the body synchrony metrics (keypoint vector scores) and the dependent variables (collective facial emotions).

We use the stepwise regression forward selection, an iterative algorithm in linear regression that optimizes the model fit by sequentially introducing predictor variables based on a predefined criterion, such as the maximization of a goodness-of-fit measure (adjusted R-squared), where each iteration adds the most relevant predictor and continues until further improvements are insignificant or a stopping condition is met.

In the SPSS tool, we choose the dependent variable as the *disgust* emotion and the predictors as 17 body synchrony scores and achieve an adjusted R squared value of 0.584. The value of N for this regression analysis is 6941. Table 2 exhibits the results of the regression analysis.

### 4.3. Deep Learning Model

The collective facial emotions of jazz musicians are predicted by utilizing body synchrony features extracted from the entanglement software.

XGBoost achieves an accuracy of 57.171% on our training data, but we favor 1D CNNs since it is particularly well suited for sequential data [24] like our time series dataset. 1D CNNs automatically learn relevant features, reducing manual feature engineering, unlike XGBoost which relies on handcrafted features.

XGBoost is an optimized gradient boosting algorithm that uses an ensemble of weak learners (typically decision trees) and applies regularization to prevent overfitting. It minimizes the sum of a specific loss function over the predictions and the gradients of the loss function to iteratively improve the model’s performance. A 1D Convolutional Neural Network (CNN) is a type of deep learning architecture designed to process one-dimensional sequential data, such as time series or sequences. It applies a one-dimensional convolution operation, utilizing filters to extract local patterns from the input data, followed by non-linear activation functions and pooling layers to learn hierarchical representations and reduce spatial dimensions.

To achieve this, a *Fully Connected Network* incorporating *1-d Convolutional Neural Network* and *Dense Layer* is employed for a multi-class classification task. 1-D CNN applies a set of learnable filters (also known as kernels) to the input features. Each filter performs a convolution operation by sliding over the input and calculating dot products between the filter and the local input patches. This process helps in capturing local patterns and features. We build a CNN-based deep learning model for predicting collective emotions using body synchrony scores. The neural network architecture is as follows. The input layer consists of the 17 body synchrony scores. The *Convolutional Neural Network (CNN)* comprises three 1-D *Convolution* layers with a kernel size of 3, filter size of 32, 64, and 64, respectively, and *ReLU* activation function. The layers are interleaved with *Max Pooling* layers of size 2. The output of these layers is flattened and fed to *Dense* layers of 64, 32, and 3 units. The activation function used for *Dense* Layers is *ReLU* except for the last layer, which employs the *Softmax* Activation. We use *Adam Optimizer* (learning rate of 0.0001) and a batch size of 100 for the dataset. The *categorical cross-entropy loss* is used for the multi-class classification. We notice saturation during training at around 17 iterations; hence, we fix the number of epochs to 20 and train the model on an NVIDIA Tesla K80 GPU. We achieve a training accuracy of 61.467% and a test accuracy of 66.168% depicted in Figure 11. Figure 12 demonstrates the confusion matrix for our deep learning model. Figure 13 displays the decrease of train and test loss as we reach saturation within 20 epochs.

## 5. Discussion

We refer to Figure 9 for interpreting the results of our correlation analysis. For the collective emotion of *disgust*, the r values of *r_knee_to_r_ank* (−0.51), *l_hip_to_l_knee* (−0.39), *r_hip_to_r_knee* (−0.35) and *l_knee_to_l_ank* (−0.35) are particularly highly negative, indicating a negative correlation between the leg movements of jazz musicians and the collective emotion of *disgust*. We also observe that the *neck_to_nose* (−0.43) and *l_eye_to_l_ear* (−0.39) values are negatively correlated with the *disgust* emotion. The movement we are looking at over here is the shaking of their head. Apart from the movement of the legs and head, we also look at the hand movements. The r values of *r_elb_to_r_wri* (−0.41), *l_elb_to_l_wri* (−0.41), *r_sho_to_r_elb* (−0.36) and *l_sho_to_l_elb* (−0.41) reveal that the hand movements of the jazz musicians are also highly negatively correlated with the emotion of *disgust*. All the above correlations have a significance value (p) of less than 0.0001. We corroborate that the musicians engaging in jazz have reduced levels of disgust and strong feelings of liking and enjoyment and have the potential to foster a state of synchronicity and flow, wherein they individually and collectively experience a harmonious alignment in their thoughts, actions, and emotions. Correlation of the *surprise* emotion is also observed to be negative having values of *r_elb_to_r_wri* (−0.44), *r_knee_to_r_ank* (−0.41), *neck_to_nose*(−0.37), *l_sho_to_r_elb*(−0.36) and *l_elb_to_l_wri*(−0.36) with a significance value (*p*) of less than 0.0001. This can be understood as an implication where being less surprised, having anticipation, prior instrument practice, and being well-rehearsed can be directly connected to being in a state of flow and synchronization. The musicians are more likely to be entangled when they collectively practice more to attain perfect synchronization. Other correlations with emotions of *sadness*, *anger*, and *fear* also prove to be negative implying that synchronization in head, arm, and leg movements among musicians indicates strong team entanglement and a state of flow with the musicians feeling more joyous.

## 6. Limitations

A key limitation of this work is that the evaluation is based on data from a relatively small group of musicians. Hence, data scarcity is one of the limitations observed. Synchronization estimation can also be improved by integrating multiple camera views. This way, occlusion issues can be prevented and synchronization scores become more robust. For the current data, we handled the occlusions by removing data points that had an obstructed camera view as well as eliminating the snippets where the musicians were not seen to be playing instruments explicitly.

## 7. Future Work and Conclusions

We presented a deep-learning approach for detecting group facial emotions leveraging body synchrony scores as features extracted from real-time multi-person pose synchronization software. We achieve a training and test accuracy of 61.467% and 66.168%, respectively. We also were able to draw conclusions about the correlations between the collective facial emotions of musicians and the synchronization between head, leg, and arm movements. Future directions include predicting human emotions using a larger dataset containing more musicians. We also envision implementing state-of-the-art deep learning models and studying different modes of data for emotion recognition like physiological signals or electrical signals generated by plants in the vicinity. This can help us discover interesting relationships between body synchrony, multi-modal data, collective emotions, and group flow. We aim to build emotion recognition models that use multiple modes of data to yield a higher accuracy while also preserving human privacy and safety. 

## Figures and Tables

**Figure 1 sensors-23-06789-f001:**
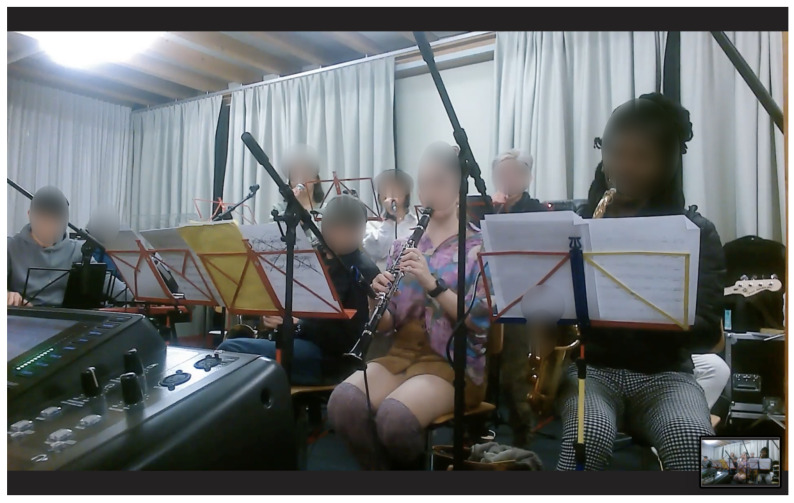
Orchestra of jazz musicians playing diverse musical instruments.

**Figure 2 sensors-23-06789-f002:**
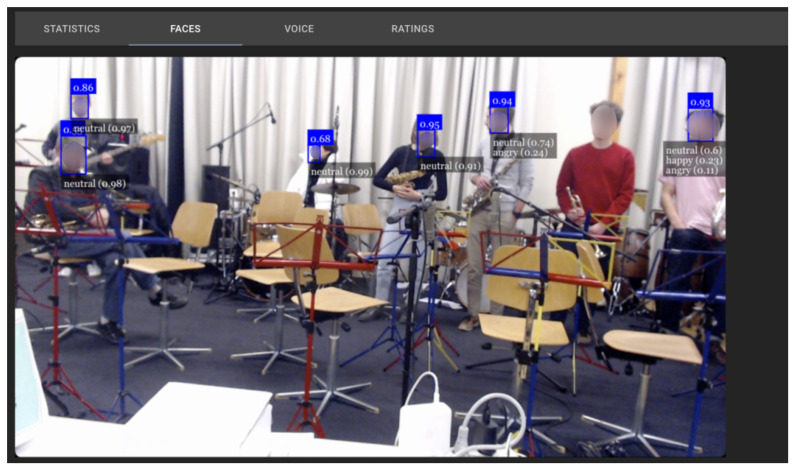
Output of the real-time group Face Emotion Recognition (FER) algorithm.

**Figure 3 sensors-23-06789-f003:**
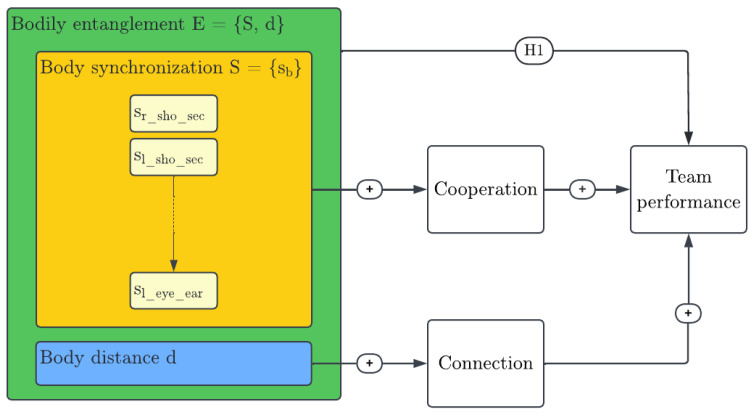
Research model for effect of bodily entanglement on team performance.

**Figure 4 sensors-23-06789-f004:**
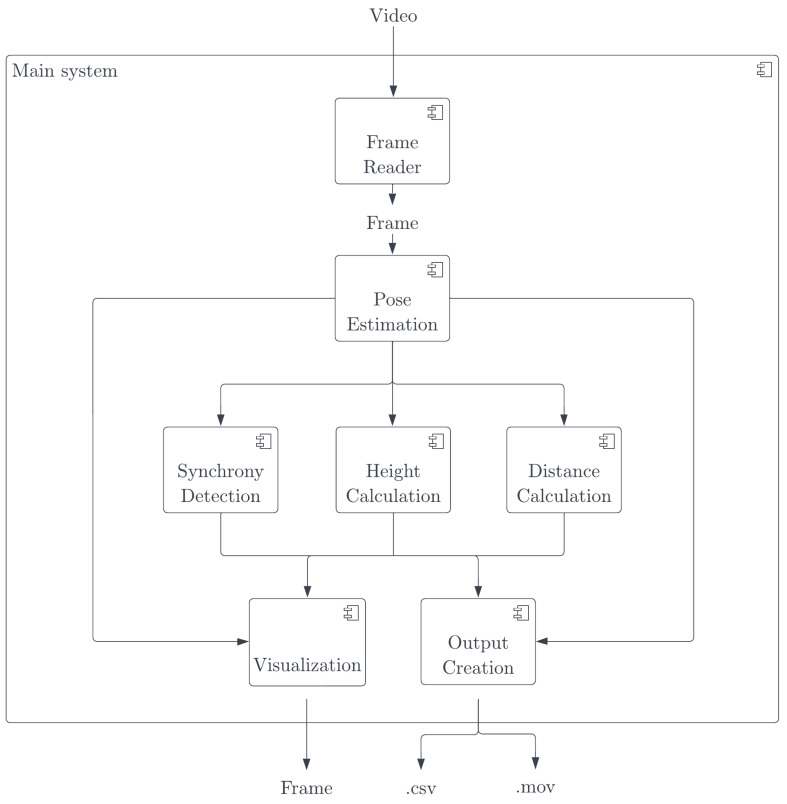
Components of the real-time multi-person pose synchronization software.

**Figure 5 sensors-23-06789-f005:**
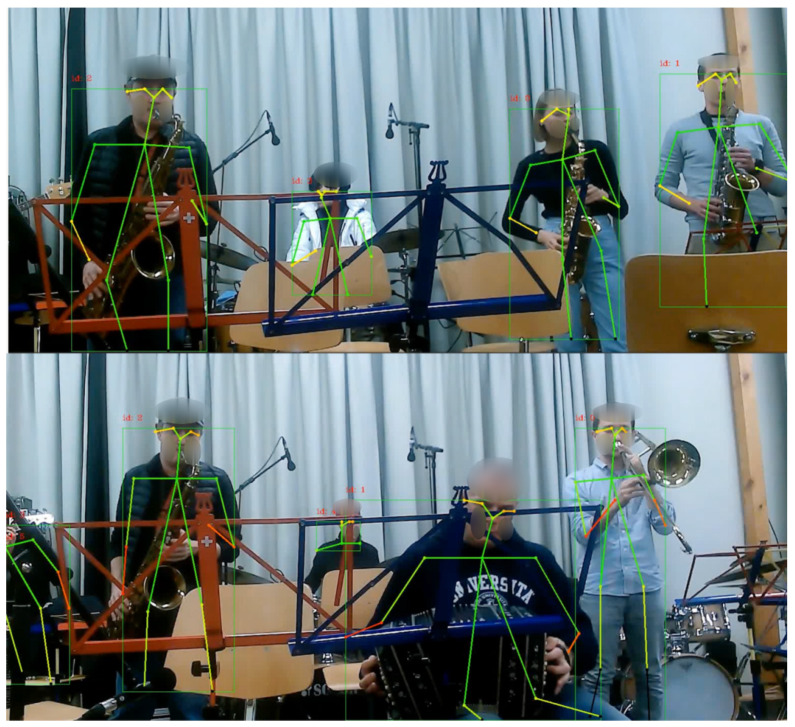
Exemplary visual output generated by our multi-person pose synchronization software.

**Figure 6 sensors-23-06789-f006:**
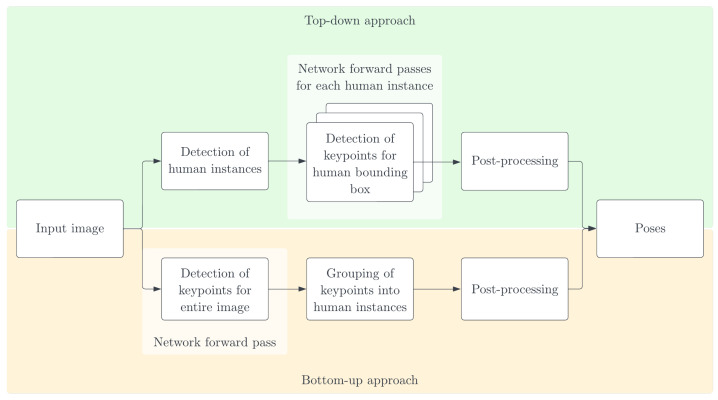
Bottom-up and top-down approaches for multi-person pose estimation.

**Figure 7 sensors-23-06789-f007:**
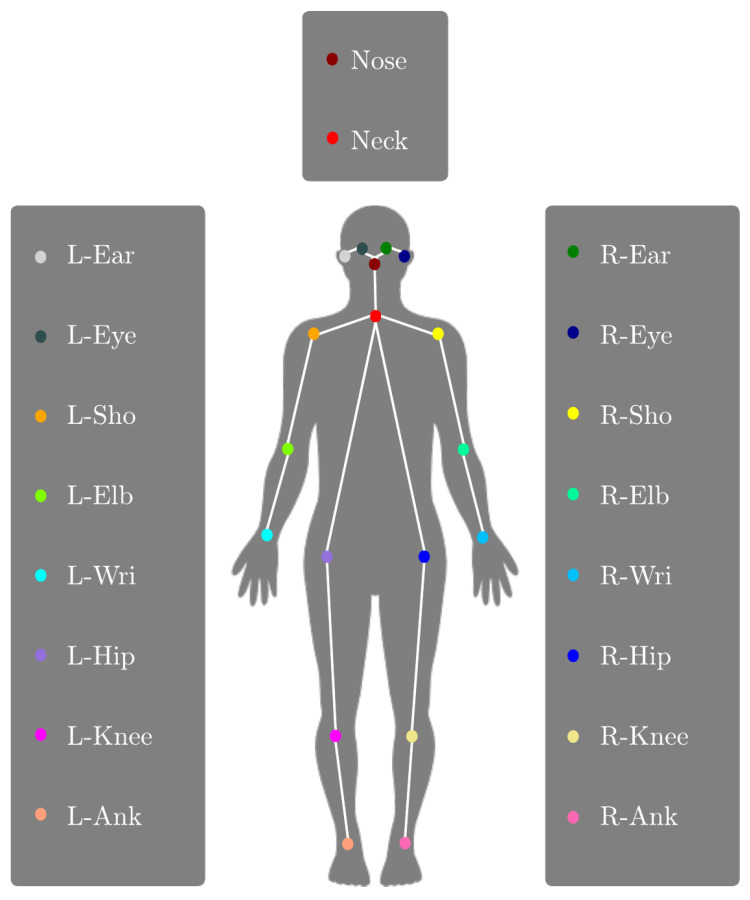
Body keypoints estimated by Lightweight OpenPose.

**Figure 8 sensors-23-06789-f008:**
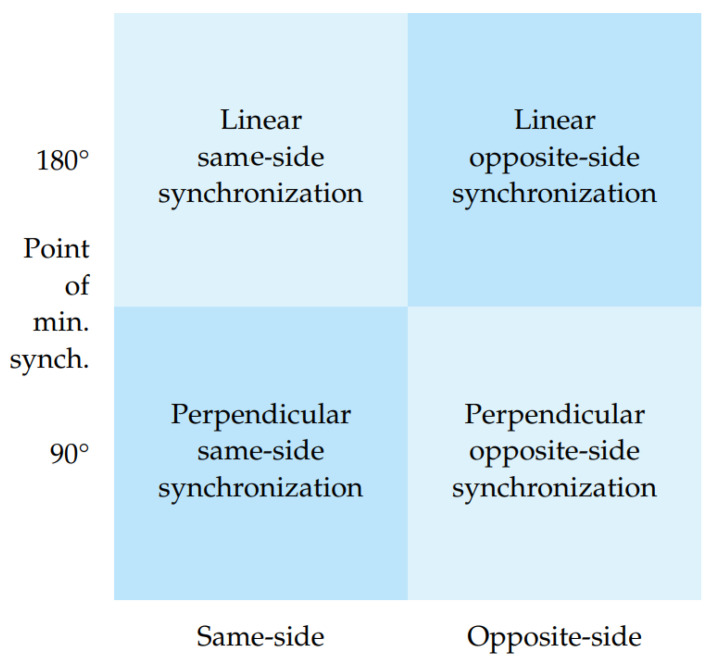
Dimensions and variants of synchronization metrics available in our software.

**Figure 9 sensors-23-06789-f009:**
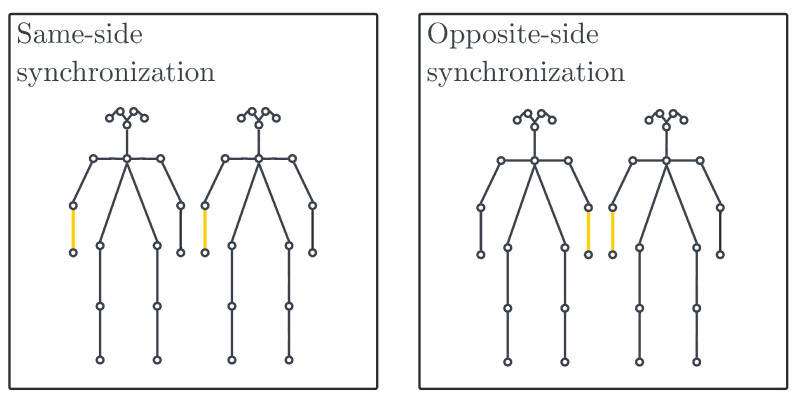
Same-side and opposite-side synchronization options.

**Figure 10 sensors-23-06789-f010:**
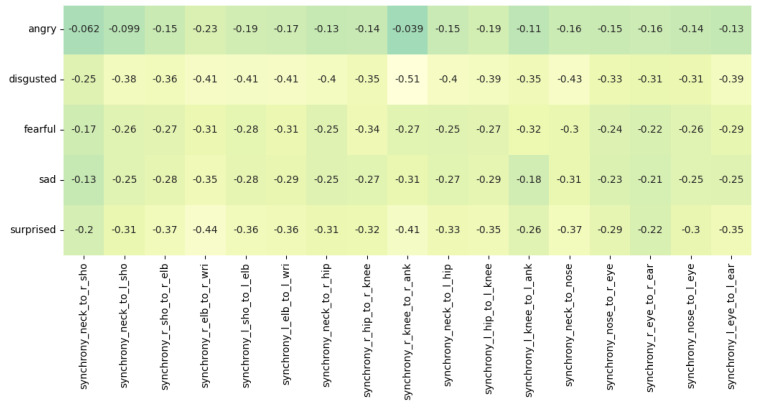
Diagonal correlation heatmap of body synchrony scores and the collective emotions of disgust, surprise, sadness, anger, and fear.

**Figure 11 sensors-23-06789-f011:**
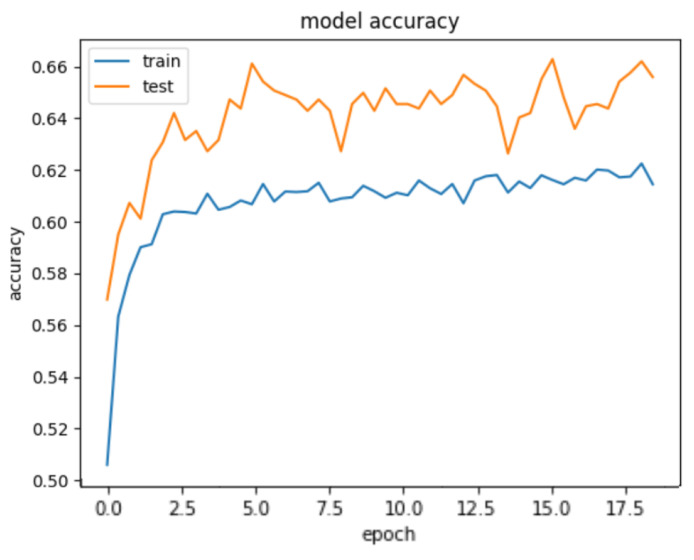
Train and test accuracy of our deep learning model.

**Figure 12 sensors-23-06789-f012:**
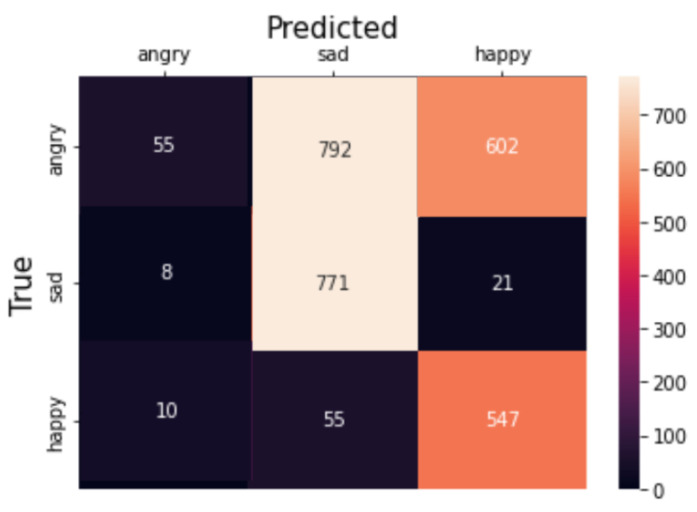
Confusion matrix of our deep learning model.

**Figure 13 sensors-23-06789-f013:**
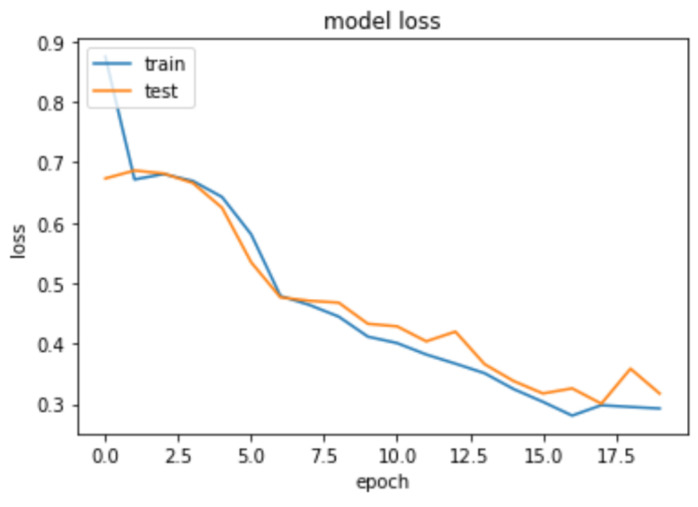
Train and test loss of our deep learning model.

**Table 1 sensors-23-06789-t001:** Body parts and corresponding keypoint vectors.

Body Part	Keypoint Vector
Right Shoulder Section	neck → r_sho
Left Shoulder Section	neck → l_sho
Right Upper Arm	r_sho → r_elb
Right Lower Arm	r_elb → r_wri
Left Upper Arm	l_sho → l_elb
Left Lower Arm	l_elb → l_wri
Right Upper Bodyline	neck → r_hip
Right Upper Leg	r_hip → r_knee
Right Lower Leg	r_knee → r_ank
Left Upper Bodyline	neck → l_hip
Left Upper Leg	l_hip → l_knee
Left Lower Leg	l_knee → l_ank
Neck Section	neck → nose
Right Nose to Eye Section	nose → r_eye
Right Eye to Ear Section	r_eye → r_ear
Left Nose to Eye Section	nose → l_eye
Left Eye to Ear Section	l_eye → l_ear

**Table 2 sensors-23-06789-t002:** Regression Analysis using SPSS (dependent variable = ’disgust’; R sq. adj = 0.584).

	Unstandardized Coefficients		Standardized Coefficients		
Model	B	Std. Error	Beta	t	Sig.
(Constant)	0.008	0.000		25.629	$5.28\times 10^{-10}$
synchrony_r_knee_to_r_ank	−0.004	0.000	−0.371	−32.068	$1.19\times 10^{-10}$
synchrony_neck_to_l_hip	0.010	0.002	0.456	5.473	$4.68\times 10^{-10}$
synchrony_neck_to_r_sho	0.033	0.001	1.451	32.711	$6.23\times 10^{-10}$
synchrony_neck_to_r_hip	−0.022	0.001	−1.118	−15.128	$2.00\times 10^{-10}$
synchrony_r_sho_to_r_elb	0.020	0.001	0.823	20.699	$7.74\times 10^{-10}$
synchrony_neck_to_nose	−0.020	0.001	−0.789	−20.238	$4.30\times 10^{-10}$
synchrony_neck_to_l_sho	−0.014	0.001	−0.695	−11.820	$9.44\times 10^{-10}$
synchrony_r_eye_to_r_ear	−0.012	0.001	−0.417	−16.156	$4.67\times 10^{-10}$
synchrony_nose_to_l_eye	0.012	0.001	0.414	10.885	$3.06\times 10^{-10}$
synchrony_l_sho_to_l_elb	−0.012	0.001	−0.567	−9.294	$2.29\times 10^{-10}$
synchrony_l_knee_to_l_ank	−0.002	0.000	−0.115	−9.702	$4.96\times 10^{-10}$
synchrony_nose_to_r_eye	0.005	0.001	0.226	5.763	$8.83\times 10^{-10}$
synchrony_r_hip_to_r_knee	0.003	0.000	0.123	5.084	$3.84\times 10^{-10}$
synchrony_l_hip_to_l_knee	−0.003	0.001	−0.136	−5.071	$4.11\times 10^{-10}$

## Data Availability

The data was collected in a private setting during the Jazzaar experiment intended solely for research purposes. The data can be made available upon confidential request for review; however, the IRB does not allow us to share it publicly.

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
