# Peer review of "Comparing Synchronicity in Body Movement among Jazz Musicians with Their Emotions"

_sensors, 2023, doi:10.3390/s23156789_

Round 1

Reviewer 1 Report

  1. Please revise the '&' symbol by text 'and' in Section 7 heading.
  2. The graphics and figures in this manuscript are of low resolution and have a small text font or text color. The authors should revise the relevant Figures.
  3. What is the significance of the last column of Table 2 because all the values in this column are zero?
  4. The sentence in line 207 is not complete; please use a dot at the end of the equation to complete it.
  5. What is the reason the data size is not equal for the correlation analysis and the regression analysis?
  6. The manuscript lacks mathematical analysis techniques.
  7. The authors need to include the importance of this study in the abstract. How can a test accuracy of 66.168% be helpful?
  8. What are the shortcomings of reasonable accuracy in this manuscript? 

Reviewer 2 Report

This paper is very well-written, with detailed and clear explanations of background, motivation, and experiment design. The exploration of the relationship between multi-person pose movement and emotion is novel and of the interest of effect computing community. The primary issue is the lack of sufficient data to support solid experiments using data-driven models, especially for neural network based models. Perhaps using xboost or a simple decision tree can achieve better performance, considering the very limited samples as pointed out by the author. The conclusion would become more convincing if more training data are harnessed. Overall, the quality of this work is high and its importance as a groundbreaking piece of research on collective emotion analysis cannot be understated. Nonetheless, addressing the issue of data scarcity could considerably amplify the impact of the research and strengthen its applicability within the field.

Reviewer 3 Report

The manuscript titled "Comparing Synchronicity in Body Movement among Jazz Musicians with their Emotions" reports a preliminary novel study investigating the relationship between the flow of a group of jazz musicians, quantified through multi-person pose synchronization, and their collective emotions. The study is a captivating and well-executed that explores a novel metric "team entanglement" that establish that higher levels of synchronized body and head movements correspond to lower levels of disgust, anger, sadness, and higher levels of joy among the musicians.

1.     Figures need better captions.

2.     For applying face-api, face landmarks should be defined. Authors should show images with clear face landmarks and bounding boxes.

3.     Line 241-245, “This is crucial as a top-down approach with a proportionally increasing computational complexity would hinder real-time analysis.” This is important fact, however, authors must elaborate.

4.     Limitation: “The accuracy of tools used for collecting facial emotions can cause our model to be less precise.” This is the biggest concern about this study. Authors should justify that why did they ignore facial recognition in team entanglement metric?

Reviewer 4 Report

This paper presents preliminary research that investigates the relationship between the flow of a group of jazz musicians, quantified through multi-person pose synchronization, and their collective emotions. It is a well-structured paper with interesting results. However, it requires further improvements.

(1) The abstract should be improved. Your point is your own work that should be further highlighted.

(2) The parameters in expressions are given and explained.

(3) The main contributions of this paper should be further summarized and clearly demonstrated. This reviewer suggests the authors exactly mention what is new compared with existing approaches and why the proposed approach is needed to be used instead of the existing methods.

(4) The values of parameters could be a complicated problem itself, how the authors give the values of parameters in the used methods.

(5) The literature review is poor in this paper. I hope that the authors can add some new references in order to improve the reviews. For example, http://dx.doi.org/10.1145/3513263; https://doi.org/10.1016/j.ins.2023.03.142; http://dx.doi.org/10.1016/j.marstruc.2022.10318; http://doi.org/10.3390/app13095706 and so on.

(6) In Section 3.3, 17 keypoint vectors is how to determine? Please provide reason.

(7)    In Figure 7, the font is too large.

(8)    In Section 4.3, why use 1-D CNN? The other deep learning models?

 Extensive editing of English language required

Round 2

Reviewer 1 Report

The authors response to my previous questions are acceptable. The authors have scope to provide descriptions of the used algorithms.

Reviewer 3 Report

This revised version of the manuscript responses reviewers' comments satisfactorily, therefore manuscript is suggested for acceptance. 

Reviewer 4 Report

OK

OK
